The (mis)measurement of the Dark Triad Dirty Dozen: exploitation at the core of the scale

Kajonius Petri J. 1 2 3 4
Persson Björn N. 3 4
Rosenberg Patricia 4 5
Garcia Danilo 1 4 5 6 danilo.garcia@icloud.com
1 Department of Psychology, University of Gothenburg , Gothenburg , Sweden
2 Department of Social Psychology, University of Skövde , Skövde , Sweden
3 Department of Cognitive Neuroscience and Philosophy, University of Skövde , Skövde , Sweden
4 Network for Empowerment and Well-Being, University of Gothenburg , Gothenburg , Sweden
5 Blekinge Center of Competence, Blekinge County Council , Karlskrona , Sweden
6 Institute of Neuroscience and Physiology, Sahlgrenska Academy , Gothenburg , Sweden
Cloninger C. Robert
Electronic publication date: 2016 Mar 1
Publication date: 2016
Volume: 4
Electronic Location ID: e1748
Received 2015 Dec 21; Accepted 2016 Feb 10
Copyright: ©2016 Kajonius et al.
Copyright year: 2016
Copyright holder: Kajonius et al.
License: This is an open access article distributed under the terms of the Creative Commons Attribution License, which permits unrestricted use, distribution, reproduction and adaptation in any medium and for any purpose provided that it is properly attributed. For attribution, the original author(s), title, publication source (PeerJ) and either DOI or URL of the article must be cited.
License URL: https://creativecommons.org/licenses/by/4.0/

Keywords: Dark triad, Dark Triad Dirty Dozen, Item response theory, Narcissism, Psychopathy, Single item dirty dark triad, Machiavellianism, Gender

Funding: The Swedish Research Council 2015-01229 The development of this article was funded by a grant from the Swedish Research Council (Dnr. 2015-01229). The funders had no role in study design, data collection and analysis, decision to publish, or preparation of the manuscript.

==============================
Background. The dark side of human character has been conceptualized in the Dark Triad Model: Machiavellianism, psychopathy, and narcissism. These three dark traits are often measured using single long instruments for each one of the traits. Nevertheless, there is a necessity of short and valid personality measures in psychological research. As an independent research group, we replicated the factor structure, convergent validity and item response for one of the most recent and widely used short measures to operationalize these malevolent traits, namely, Jonason’s Dark Triad Dirty Dozen. We aimed to expand the understanding of what the Dirty Dozen really captures because the mixed results on construct validity in previous research.

Method. We used the largest sample to date to respond to the Dirty Dozen (N = 3,698). We firstly investigated the factor structure using Confirmatory Factor Analysis and an exploratory distribution analysis of the items in the Dirty Dozen. Secondly, using a sub-sample (n = 500) and correlation analyses, we investigated the Dirty Dozen dark traits convergent validity to Machiavellianism measured by the Mach-IV, psychopathy measured by Eysenck’s Personality Questionnaire Revised, narcissism using the Narcissism Personality Inventory, and both neuroticism and extraversion from the Eysenck’s questionnaire. Finally, besides these Classic Test Theory analyses, we analyzed the responses for each Dirty Dozen item using Item Response Theory (IRT).

Results. The results confirmed previous findings of a bi-factor model fit: one latent core dark trait and three dark traits. All three Dirty Dozen traits had a striking bi-modal distribution, which might indicate unconcealed social undesirability with the items. The three Dirty Dozen traits did converge too, although not strongly, with the contiguous single Dark Triad scales (r between .41 and .49). The probabilities of filling out steps on the Dirty Dozen narcissism-items were much higher than on the Dirty Dozen items for Machiavellianism and psychopathy. Overall, the Dirty Dozen instrument delivered the most predictive value with persons with average and high Dark Triad traits (theta > −0.5). Moreover, the Dirty Dozen scale was better conceptualized as a combined Machiavellianism-psychopathy factor, not narcissism, and is well captured with item 4: ‘I tend to exploit others towards my own end.’

Conclusion. The Dirty Dozen showed a consistent factor structure, a relatively convergent validity similar to that found in earlier studies. Narcissism measured using the Dirty Dozen, however, did not contribute with information to the core of the Dirty Dozen construct. More importantly, the results imply that the core of the Dirty Dozen scale, a manipulative and anti-social trait, can be measured by a Single Item Dirty Dark Dyad (SIDDD).

“Show me again, the power of the darkness, and I’ll let nothing stand in our way. Show me, grandfather, and I will finish what you started.”

From Star Wars: The Force Awakens

Over the last 25 years, the vast majority of personality research has focused on the Big Five traits: openness, conscientiousness, extraversion, agreeableness, and neuroticism. The Big Five Model of personality is a theory developed from both language taxonomy as well as statistical factor analysis (Costa & McCrae, 1992). Critics have argued against what they believe is the overreliance on factor analysis (i.e., one of the methods in Classical Test Theory) to uncover the latent structure of personality, without a well-grounded theoretical basis and substantial variation in methodology (e.g., Block, 1995; Gould, 1981). In the last decade, personality psychologists have turned their attention to the dark side of human character: Machiavellianism, psychopathy, and narcissism. Together, these traits are widely known as the Dark Triad model (Paulhus & Williams, 2002). The validation studies of Dark Triad measures have mostly been conducted using Classic Test Theory methods and in very few cases using Item Response Theory (IRT) methods. Generally speaking, the Dark Triad embodies interpersonal, sub-clinical, and maladaptive personality traits in the general population (Paulhus & Williams, 2002), which are characterized by manipulativeness (i.e., Machiavellianism), impulsivity and antagonism (i.e., psychopathy), and the sense of entitlement (i.e., narcissism). The Dark Triad traits are associated with a value system of unconventional and antisocial morality (Kajonius, Persson & Jonason, 2015; for a more extensive review see Furnham, Richards & Paulhus, 2013; Paulhus, 2014). In essence, individuals with high levels on any of these dark traits appear to operate in selfish and competitive ways with a common core of uncooperativeness (see Jones & Figueredo, 2013). Thus, whether the dark traits constitute a ternary model of unique traits or a unified uncooperative general factor with three closely related anti-social sub-traits is still a question for research. In this context, validation studies using IRT methods might shed some light on what different measures of the Dark Triad actually measure.

As with most personality psychology research, the measurement of individuals’ tendencies on the dark traits is often conducted using self-report measures. Most of the time, this has been done using one instrument for each trait. These single instruments are often long and time demanding. For the trait of Machiavellianism, for example, researchers often use Christie and Geis’ Mach-IV (1970), which was originally based on statements from the Italian Niccolò Machiavelli’s books The Prince and The Discourse (see also Jones & Paulhus, (2009), who point out that the instrument also captures behaviors from the Chinese military general, strategist, and philosopher Sun Tzu’s book The Art of War; behaviors such as planning, building a reputation, and creating alliances). The trait of psychopathy is often measured with the Self-report Psychopathy Scale (Hare, 1985). This instrument was first used on prisoners and later on validated in non-criminal populations as well (Hare, 1985). Nevertheless, also the psychoticism scale in the hierarchical three-factor model proposed by Eysenck (e.g., Eysenck, Eysenck & Barrett, 1985) has, even though not without criticism, been used as a measure of psychopathy in terms of “Impulsive Unsocialized Sensation Seeking” (Zuckerman et al., 1991. See also Zuckerman, 1989; Zuckerman, 1991; Linton & Power, 2013; Garcia & Sikström, 2014). Finally, narcissism is often measured using the Narcissism Personality Inventory, which comprises 80 (long version) or 32 (short version) paired-items (Raskin & Hall, 1979). More recently, shorter measures comprising all three traits in one single instrument have been created to facilitate data collection.

One such measure is the Dark Triad Dirty Dozen (Jonason & Webster, 2010). See Table 1 for the statements and the keyword in each one of the statements in the Dirty Dozen scale. The Dirty Dozen comprises 12 items that in four consecutive studies were demonstrated to retain its core of disagreeableness when compared to 91 items from questionnaires that measured the dark traits separately (Jonason & Webster, 2010). This is a reduced item count by 87%. Subsequent studies with smaller samples have explored the thin line between efficiency and accuracy in this short scale. The findings suggest that a bi-factor structural model with both a general latent Dark Triad construct and the three dark traits of Machiavellianism, psychopathy, and narcissism best fit the data. In addition, the findings also show relatively good convergent validity with the Mach-IV (r = .53), the Self-report Psychopathy Scale III (r = .32), and the Narcissism Personality Inventory-40 (r = .53) (Jonason & Luévano, 2013). Further validations, using a sample of young undergraduates, were reported with the ubiquitous Big Five Inventory developed by Benet-Martínez & John (1998). The findings revealed an unstable core of conscientiousness for psychopathy and agreeableness for both Machiavellianism and psychopathy, with no clear relationships with extraversion for narcissism (Jonason et al., 2013). Again, a bi-factor model (i.e., one general factor plus three specific factors) fitted the data best. This suggests that each dark trait measured something unique (Jonason et al., 2013), in addition to the common variance captured by the general factor. However, criticism has also been leveled against the Dirty Dozen. For instance, the Dirty Dozen’s incremental, discriminant, and convergent validity has been called into question when compared with other relevant measures (Jones & Paulhus, 2014; Maples, Lamkin & Miller, 2014; Miller et al., 2012). Additionally, its construct validity has been disputed, as using merely 4 items/factor may remove essential content (Miller et al., 2012). In addition to Classical Test Theory current research for validation of Dark Triad scales has used IRT models.

Table 1 Jonason’s Dark Triad Dirty Dozen Scale: traits, item numbers, statements, and keyword in each one of the statements.

Trait	Item No.	Statement	Keyword	
Machiavellianism	1	I tend to manipulate others to get my way.	Manipulate	
2	I have used deceit or lied to get my way.	Deceit	
3	I have use flattery to get my way.	Flatter	
4	I tend to exploit others towards my own end.	Exploit	
Psychopathy	5	I tend to lack remorse.	Remorse	
6	I tend to be unconcerned with the morality of my actions.	Amoral	
7	I tend to be callous or insensitive.	Callous	
8	I tend to be cynical.	Cynical	
Narcissism	9	I tend to want others to admire me.	Admire	
10	I tend to want others to pay attention to me.	Attend	
11	I tend to seek prestige or status.	Status	
12	I tend to expect special favors from others.	Favors	
Notes.

Adapted from Jonason & Webster (2010).

There is a large diversity of models that have been developed using IRT. IRT was first proposed in the field of psychometrics for the purpose of ability assessment. For instance, all major educational tests are developed using this technique because it significantly improves measurement accuracy and reliability, and it provides significant reductions in assessment time and effort (for a review see An & Yung, 2014). In recent years, this technique has also been applied in health and clinical research (e.g., Hays, Morales & Reise, 2000; Edelen & Reeve, 2007; Holman, Glas & De Haan, 2003; Reise & Waller, 2009). Using IRT models, researchers have found a slightly lower endorsement threshold of the dark traits for males compared to females. This has been interpreted as differences in social undesirability sensitivity, or true differences as proposed by mating-strategy theory (Webster & Jonason, 2013). The latest validation study among onsite UK undergraduates and online Crowdflower-workers,1 however, found conflicting results using Mokken analysis, a non-parametric form of IRT (Carter et al., 2015). While the expected three traits of Machiavellianism, psychopathy, and narcissism, emerged among female students’ scores; only two traits emerged among male students’ scores. These two traits were a combined Machiavellianism-psychopathy factor and a narcissism factor. In contrast, among the online workers, only one core construct of the Dirty Dozen appeared. These differences were not explained by invariance over sex and age. Hence, this casts some uncertainty on the evasive constructs measured by the Dirty Dozen scale or suggests some kind of mismeasurement of the triad by this specific scale.

The present study

The possibility to replicate findings is one of the parameters that distinguish science from non-science. In short, replication should be at “the heart of science” (Schmidt, 2009). By use of conceptual replications we can potentially confirm which findings about human nature that can be generalized and thus increase predictive validity in our regular use of psychological measurements. As researchers we expect that replication studies are common and that the methodology is well developed, however, particularly in social sciences the contrary is true, demonstrating an overall replication rate of only 1.07% (Makel, Plucker & Hegarty, 2012; see also Lucas & Donnellan, 2013, and the Registered Replication Reports initiative by the Association for Psychological Science, http://www.psychologicalscience.org/index.php/replication).

A major problem in current validations is the small sample sizes and a general lack of power and precision. This “results in lower precision in parameter estimates and systematically inflated effect size estimates” (Lucas & Donnellan, 2013, p. 453). In addition, the current validation studies that have been published provide many statistically significant low-powered findings even within the same study, which “paradoxically provide less support for a phenomenon than papers that report some failures to reach statistical significance” (Lucas & Donnellan, 2013, p. 453; see also Francis, 2012; Schimmack, 2012). To the best of our knowledge, the present study provides the largest single sample used to this date (N = 3, 698) to replicate some of the most common findings with regard to the Dirty Dozen scale. For instance, previous validation studies on the Dirty Dozen have had limited, or at least unclear, generalizability, often only including undergraduates or homogenous age cohorts. In addition, although we do believe in researchers’ capacity for objectivity, we see as an important venue that an independent research group that had no ties to the construction of the Dirty Dozen scale conducted the present replication study.

First, we used Confirmative Factor Analysis (CFA) to investigate the original factor structure of the Dirty Dozen, which has shown varied results in previous studies, this time using a sizable, heterogeneous sample from all walks and ages of life. In addition, with the large sample at hand, there were sufficient respondents for an exploratory distribution analysis, which has not been reported before. Second, we further establish the validity of the traits measured using the Dirty Dozen by investigating convergence with known, contiguous single long scales of the dark traits: the Mach-IV, Eysenck’s Personality Questionnaire Revised, and the Narcissism Personality Inventory. Third, using IRT, we explored what the Dirty Dozen endeavors to measure. We propose that what makes the Dark Triad, measured by this specific scale, “dark” may not be a uniform stable core but, instead, a challenging mix of malevolent and tradition-laden anti-social and uncooperative traits. If so, we might be able to clarify what the Dirty Dozen measures and uncover what scale-items might be responsible for the many interpretations of its core.

Method

Ethical statement

After consulting with the Network for Empowerment and Well-Being’s Review Board we arrived at the conclusion that the design of the present study (e.g., all participants’ data were anonymous and will not be used for commercial or other non-scientific purposes) required only informed consent from the participants.

Participants

The participant data was collected through Mechanical Turk (MTurk), which has demonstrated reliability and validity, providing a wider range of socio-economic backgrounds compared to other samples (Casler, Bickel & Hackett, 2013). This is particularly useful when it comes to research on values, such as the undesirability of the Dark Triad traits’ values (cf. Kajonius, Persson & Jonason, 2015). All participants were informed that the survey was voluntary, anonymous, and that the participants could terminate the survey at any time. The MTurk workers received 50 cents (US-dollars) as compensation for participating and only residents of the US were allowed to accept participation. Two control questions were added to the survey, to control for automatic responses (e.g., “This is a control question, please answer “neither agree or disagree”). A total of 50 participants responded erroneously to one or both of the control questions, the final sample constituted 3,698 (Mage = 33.5, SD = 11.8). As expected, males (Nmales = 1,726) scored higher on all Dark Triad traits than females (Nfemales = 1,972), as summarized in the descriptive Table 2. A subsample (N = 500) also answered to single long instruments of the Dark Triad, extraversion, and neuroticism.

Table 2 Descriptive Analysis of the Dark Triad traits as measured by the Dark Triad Dirty Dozen.

	M	SD	α	Skewness	Kurtosis	Mmale	SDmale	αmale	Mfemale	SDfemale	αfemale	
1 DD Machiavellianism	3.00	1.41	0.80	0.38	−0.66	3.23	1.45	.84	2.79	1.35	.85	
2 DD Psychopathy	2.42	1.26	0.76	0.95	0.43	2.74	1.30	.81	2.13	1.14	.79	
3 DD Narcissism	3.55	1.44	0.81	−0.15	−0.81	3.71	1.43	.75	3.41	1.44	.74	
4 Dark Triada	2.99	1.08	0.85	0.26	−0.27	3.23	1.08	.80	2.78	1.04	.81	
Notes.

N = 3,698; Nmales = 1,726; Nfemales = 1,972

a Composite score of the three dark traits.

Measures

The Dark Triad Dirty Dozen (Jonason & Webster, 2010) is a 12-item self-report questionnaire measurement of the three Dark Triad traits. Participants are asked to rate how much they agreed (1 = Strongly disagree; 7 = Strongly agree) with statements such as: “I tend to manipulate others to get my way” (Machiavellianism), “I tend to lack remorse” (psychopathy), and “I tend to want others to admire me” (narcissism). Items were averaged to create each dimension (Cronbach’s alphas between .74 and .85; see Table 2 for alphas for both males and females). We also constructed a composite score of the three dark traits by using the mean values from all of the items.2 For facilitating readability, all measures of the dark traits using the Dirty Dozen are labeled as follows: DD Machiavellianism, DD psychopathy, and DD narcissism. High scores represent high degree in each of the dark traits or, in the case of the composite, a high degree of the Dark Triad Dirty Dozen core.

The Mach-IV (Christie & Geis, 1970) was used to also measure Machiavellianism. The Mach-IV consists of 20 items that reflect ways of thinking and opinions about people and different situations (e.g., “Never tell anyone the real reason you did something unless it is useful to do so”). Participants were requested to rate to what extent they agree with each statement on a 6-point Likert scale: 1 = Strongly agree, 6 = Strongly disagree. The Machiavellianism score was computed by summarizing the means across the 20 items, a high score representing high degree of Machiavellianism.

The short version of the Eysenck’s Personality Questionnaire Revised was used to measure extraversion (e.g., “Do you usually take the initiative in making new friends?”), neuroticism (e.g., “Do you ever feel ‘just miserable’ for no reason?”), and psychoticism (e.g., “Would you like other people to be afraid of you?”) (Eysenck, Eysenck & Barrett, 1985). The Eysenck questionnaire consists of 12 items for each trait (forced binary answers: Yes or No). The score for each of the personality traits was computed as the sum of the 12 items, with yes responses coded as 1 and no responses coded as 0. Thus, a high score represents high degree in each of the three personality traits. As stated in the Introduction section, Eysenck’s psychoticism scale has, however imperfectly, been suggested as partly tapping into the sensation-seeking/boldness part of psychopathy (cf. Zuckerman, 1989; Zuckerman, 1991). For the rest of the paper we refer to the psychoticism scale as psychopathy.

The short version of the Narcissistic Personality Inventory was used to also measure Narcissism (Ames, Rose & Anderson, 2006). The instrument consists of 16 pairs of items (one consistent and one inconsistent with narcissistic behavior in each pair) for what participants are instructed to choose, for each pair, one item that comes closest to describing their own feelings and beliefs about themselves. The narcissism score was computed as the sum of the 16 items, with narcissism-consistent responses (e.g., “I really like to be the center of attention”) coded as 1 and narcissism-inconsistent responses coded as 0 (e.g., “It makes me uncomfortable to be the center of attention”). Thus, a high score represents high degree of narcissism.

Statistical analysis

As in earlier studies, there was a relatively large skewness in the psychopathy scores and kurtosis in the narcissism scores. This has, however, been shown to not have a negative effect on subsequent statistical analysis when the sample size reaches the thousands (Lumley et al., 2002). First, using Classic Test Theory, we used CFA for testing two contending models, one with only the latent dark triad core branching into three dark traits, and second, a bi-factor model with the latent dark triad core connecting directly with all items, while the three dark traits connecting only to their respective items. We used Structural Equation Modeling (SEM) in the software Amos version 22 for these calculations. In addition, we conducted an exploratory distribution analysis using the large sample at hand. Second, using the subsample, we conducted convergent correlational analyses in SPSS version 22 with the collected contiguous single dark traits scales and Extraversion and Neuroticism (i.e., Mach-IV, Eysenck’s Personality Questionnaire Revised, and the Narcissism Personality Inventory). Third, using the large sample and with the purpose of exploring the Dirty Dozen content, we utilized the much in-demand method of IRT using the R package MIRT version 1.10 (Chalmers, 2012) in R version 3.2.1 (R Core Team, 2015). This is a methodology for modeling how test items contribute to one latent, scalable trait. We used a graded response model analogous to the two-parameter model for dichotomous items, which basically generates two defining characteristics for each item: a slope coefficient, or discrimination parameter alpha (a), and a discrimination coefficient, or threshold parameter beta (b). The a parameter shows how strongly an item relates to a given latent construct theta (θ; which in this study the Dark Triad core as measured by the Dirty Dozen scale). The a parameter can be analogized as a factor loading, whereas the threshold parameters b1--6 relates to the level of the latent trait at which the next highest response category has at least 50% probability of being endorsed. For more information about IRT see Morizot, Ainsworth & Reise (2007).

Results

The first purpose was to replicate the original factor-structure of the Dirty Dozen using Classic Test Theory. Two CFA-models were tested. The first model, a hierarchical structure, with the Dark Triad core “above” the three dark traits, DD Machiavellianism (λ = .75), DD psychopathy (λ = .86), and DD narcissism (λ = .56), was not optimal (χ2(40) = 1530.24, p < .001) and with non-satisfactory fit indices as well (NFI = .92, CFI = .92, and RMSEA = .10). The second model tested was a bi-factor structure, which proved more successful (χ2(28) = 360.19, p < .01) with sufficient fit indices (NFI = .98, CFI = .98, and RMSEA = .05). The RMSEA of this specific model was slightly better than in previous studies (RMSEA = .07 in Jonason & Luévano, 2013; RMSEA = .06 in Jonason et al., 2013). Furthermore, our model showed that 3 out of the 4 items in the DD narcissism cluster had very weak relationships with the Dark Triad core and that the DD Machiavellianism-items demonstrated the strongest relationships. The full model with all items’ regression coefficients is reported in Fig. 1.

Figure 1 Bi-factor model of Dirty Dozen.

N = 3,698. NFI = .98, CFI = .98, RMSEA = .05.

In addition, with the large sample at hand, there were sufficient respondents for an exploratory distribution analysis, which as far as we know has not been reported before. Figure 2 depicts a strong bimodality of the distribution (peaks on both Likert-categories 1 and 5) found in all three dark traits. DD Psychopathy showed the largest overrepresentation in the lowest scale-category (Likert-category 1), followed by DD Machiavellianism and last DD narcissism. Females were overrepresented in the lowest scale-category (Likert-category 1) compared to males, Nfemales = 280, Nmales = 151 (DD Machiavellianism), Nfemales = 486, Nmales = 280 (DD psychopathy), and Nfemales = 188, Nmales = 114 (DD narcissism). In small sample-studies, distributions such as these, might strongly affect statistical validity, as well as external validity, indicating strong social undesirability with the items.

Figure 2 Frequency distributions showing the bi-modality of the three Dirty Dozen Dark Triad traits.

N = 3,698. The numbers on the y-axis represent the proportion of replies for each Likert-category (1–7) on the x-axis. For instance, 47% of total replies on DD psychopathy items were placed on the lowest option (1) “strongly disagree,” which depicts the skewness in response pattern.

The second purpose was to analyze convergent validity of the dark traits as measured with the Dirty Dozen scale. We were simply looking for the expected, conjoining relationships between the Dark Triad traits and the contiguous, single long scales of the dark traits and both neuroticism and extraversion. Table 3 summarizes the correlations found, which overall did show relatively weak (all rs < .50) converging relationships (Machiavellianism r = .49; psychopathy r = .41, narcissism r = .47). That is, the three dark traits measured with Dirty Dozen showed that DD Machiavellianism and DD psychopathy showed similar correlations, while DD narcissism related less well with the corresponding scales measured using the single long scales. Additionally, the Dirty Dozen Dark Triad composite (Table 3, row 4) showed smaller correlations with the dark traits measured using the single long scales (correlations between the Dirty Dozen dark traits and the Dark Triad core were between .75 and .88, correlations between the dark traits measured with the single instruments and the Dark Triad core were between .31 and 53). In addition, there are some discrepancies between the correlations between neuroticism and extraversion and the dark traits depending on how the Dark Triad was measured. For example, while there was a weak significantly positive correlation between DD narcissism and neuroticism (r = .20, p < .01), there was no significant correlation to neuroticism when narcissism was measured using the Narcissism Personality Inventory (r = − .07). In contrast, the relationship between narcissism and extraversion was almost twice as large when narcissism was measured using the single instrument (r = .40, p < .01) than when measured using the Dirty Dozen (r = .25, p < .01).

Table 3 Convergent analysis (Persons’ r) of the Dark Triad Dirty Dozen traits and the dark traits and Extraversion, and Neuroticism as measured by the Mach-IV, Eysenck’s Personality Questionnaire Revised, and the Neuroticism Personality Inventory.

	1	2	3	4	5	6	7	8	9	Item 4: exploit	
1 DD Machiavellianism	–	.58	.50	.88	.49	.27	.37	.26	.10	.80	
2 DD Psychopathy		–	.27	.75	.57	.41	.30	.25	−.11	.60	
3 DD Narcissism			–	.76	.22	.09	.47	.20	.25	.43	
4 Dark Triada				–	.53	.31	.49	.30	.11	.77	
5 Machiavellianism					–	.40	.34	.28	−.06	.50	
6 Psychopathy						–	.35	.05	.05	.30	
7 Narcissism							–	−.07	.40	.40	
8 Neuroticism								–	−.24	.18	
9 Extraversion									–	.09	
Item 4: exploit										–	
Notes.

N = 500. All r coefficients > .12 are significant at p < .01.

a Summarized composite score of the three dark traits.

Yellow fields: intra-relationships within the Dark Triad Dirty Dozen traits and the Dark Triad composite.

Blue fields: intra-relationships within the Dark Triad traits measured by the single instruments (i.e., MACH-IV, Eysenck’s Personality Questionnaire Revised, and the Narcissism Personality Inventory).

Green fields: relationships between corresponding dark traits measured using the Dirty Dozen scale and the single instruments.

Black fields: relationships between dark traits as measured by the Dirty Dozen and Neuroticism and Extraversion.

Grey fields: relationships between dark traits as measured by the single instruments and Neuroticism and Extraversion.

The third and last purpose was to extend the discussion on the construct of the Dark Triad as measured by the Dirty Dozen scale, using IRT. The results from both the CFA model 1 (NFI = .92) and 2 (NFI = .98) indicated adequate unidimensionality, which is the basic assumption for IRT. We ran a polytomous graded-response model on the 12 Dirty Dozen items, allowing items to load on a latent Dark Triad core. The Total Information Curve reported in Fig. 3 shows that the core of the Dark Triad (θ) was revealed in a maximized way only when a participant has close to average levels (−0.5) of this latent trait (see Fig. 4 for the Information Curve for each one of the dark traits measured using the Dirty Dozen). Hence, the Dirty Dozen scale functions well for capturing average and higher levels of the core Dark Triad, but not the lower levels. This once again leads to the question what constitutes the dark core, or more specifically to what the Dirty Dozen scale actually measures. See Fig. 5 for Scale Information Curves and Table 4 for Item Response Theory Rank and Exploratory Factor Analysis of the Dirty Dozen Items.

Figure 3 Total information curve on the latent core of the dark triad dirty dozen.

Overall, the Dirty Dozen instrument delivers the most predictive value with persons with average and high Dark Triad traits (theta > −0.5).

Figure 4 Scale information curves depicting the information content in each respective sub factor.

Figure 5 Item information curves for each of the 12 items of the Dirty Dozen.

Note that item 1 (Manipulate) and 4 (Exploit) deliver the most information on the latent Dark Triad (labeled theta). Item 8 (Cynicism) delivers the least information. Furthermore, the three first items on narcissism (9–11) don’t deliver much predictive information (flat curves, cf. earlier CFA bi-factor model) to the overall Dark Triad trait—However, an IRT with only the narcissism items confirms that these predict the latent trait (theta, i.e., narcissism) very satisfactorily (cf. earlier CFA hierarchical model).

Table 4 Item response theory rank and exploratory factor analysis of dirty dozen items.

		Males	Females	
	Info rank	Machiavellianism	Psychopathy	Narcissism	Machiavellianism	Psychopathy	Narcissism	
1 Manipulate*	2	.81	.54	.34	.77	.50	.38	
2 Deceit	3	.73	.46	.30	.76	.40	.35	
3 Flatter	8	.59	.27	.44	.60	.26	.43	
4 Exploit*	1	.78	.60	.36	.70	.60	.35	
5 Remorse	5	.44	.78	.08	.42	.84	.17	
6 Amoral	4	.46	.72	.15	.41	.76	.16	
7 Callous*	6	.47	.77	.13	.50	.70	.20	
8 Cynical*	11	.36	.40	.14	.45	.42	.25	
9 Admire	10	.35	.09	.81	.40	.11	.81	
10 Attend	12	.33	.08	.78	.34	.11	.80	
11 Status	9	.38	.16	.73	.41	.23	.73	
12 Favors*	7	.52	.36	.55	.51	.43	.57	
Notes.

* Potentially conflicting items in regards to double-loadings.

Bold figures, Loadings according to three-factor Dark Triad theory. Cursive figures, Loadings deviating from three-factor Dark Triad theory (>.50). Item 8, underscored, “I tend to be cynical” does not load well enough on the psychopathy-factor and deviates from theory by aligning also with Machiavellianism (especially for females). Items 1, “I tend to manipulate others to get my way,” 4, “I tend to exploit others toward my own end,” and 12, “I tend to expect special favors from others” double-load on two separate factors. Items 4, 5, and 7 have somewhat varying loadings for males and females.

In Table 5 the items’ ability to differentiate (a parameter) between people with similar levels of the same latent trait are ranked, starting with the item yielding the most information (item 4: Exploit). The a-parameter typically ranges from 0.5 to 2.0 in personality scales (Morizot, Ainsworth & Reise, 2007). As can be seen, three of the four DD narcissism items (item 10: Attend, item 9: Admire, and item 11: Status) contributed least to differentiation between individuals. The difficulties (b) for each item are listed in rows, and reflect the threshold levels of the latent trait necessary to have at least 50% chance of endorsing the next scale-step (e.g., b1 denotes answering Option 1 vs. 2, 3, 4, 5, 6, 7). The b-parameters are scaled on the same metric as the latent trait (θ) and falls in the range of −3–+3 SD, thus 0 is approximated to be of average difficulty (at the mid-point of the distribution). At the highest scale-step (b6), all sub-factor items showed extreme difficulty, close to 3 SD. At the lower end (b1), the items on DD Machiavellianism (items 1–4) and DD psychopathy (items 5–8), still showed much difficulty, close to average 0, while narcissism (items 9–12), showed much less difficulty, close to −2SD. These results imply that DD narcissism is not contributing as much information to the core constitution of the dark triad construct, since the probability of filling out steps on the DD narcissism-items are much higher than on the DD Machiavellianism and DD psychopathy-items (cf. first, the skewness in distributions in Fig. 2 and second, that DD narcissism correlated the least with the contiguous scales in Table 3). In other words, when a respondent does fill out high numbers on scale-items on DD Machiavellianism and DD psychopathy, this rapidly predicts the latent level of participant’s core dark personality (e.g., item 1: Manipulate, a = 2.73 or item 6: Amoral, a = 1.91), but not for DD narcissism (e.g., item 10: Attend, a = .91). The separate item information curves are found in Fig. 4.

Table 5 Item response theory analysis of the Dirty Dozen.

Item	a	b1	b2	b3	b4	b5	b6	
4 Exploit	3.33	−0.03	0.63	0.97	1.31	1.99	2.58	
1 Manipulate	2.73	−0.24	0.42	0.75	0.94	1.78	2.54	
2 Deceit	1.96	−0.71	−0.07	0.27	0.44	1.24	2.35	
6 Amoral	1.91	0.36	1.11	1.56	1.87	2.44	3.14	
5 Remorse	1.84	0.25	1.00	1.36	1.68	2.29	3.10	
7 Callous	1.82	−0.03	0.75	1.15	1.45	2.18	3.05	
12 Favors	1.71	−0.47	0.42	0.82	1.37	2.21	3.35	
3 Flatter	1.41	−1.43	−0.79	−0.35	−0.07	1.00	2.46	
11 Status	1.19	−1.23	−0.45	0.02	0.52	1.59	2.76	
8 Cynical	1.07	−1.13	−0.41	0.05	0.43	1.41	2.77	
9 Admire	1.02	−2.12	−1.38	−0.89	−0.18	1.07	2.83	
10 Attend	0.91	−1.99	−1.12	−0.54	0.21	1.72	3.75	
Notes.

Items are ranked according to item’s ability to discriminate (a) levels of the latent trait (the core of the Dark Triad) and are numbered according to their positions in the original questionnaire (DD Machiavellianism, 1–4, DD psychopathy, 5–8, and DD narcissism, 9–12). b1--6 reports the item difficulties, reflecting the threshold level (−3–+3 SD) of the latent trait necessary to have at least a 50% chance of endorsing the next scale-steps.

The item with the highest a (item 4: Exploit, a = 3.33), was of particular interest, due to its superior discriminatory ability compared to the others. We surmised that this item in itself would be able to capture the entire Dark Triad core, as measured by the Dirty Dozen scale. This exploitation-item correlated with the summed Dark Triad (r = .77), to the same degree that the three dark traits did, DD Machiavellianism (r = .88), DD psychopathy (r = .75), and DD narcissism (r = .76). When exchanging the summed Dark Triad for the single exploit-item, the internal reliability between the constructs was only marginally lowered: inter-item .61–.51; Cronbach’s alpha .85–.80. Finally, comparing the single item with the convergence coefficients of the summed Dark Triad composite (Table 3), the single item performed as well or better as a substitute.

Discussion

This was a replication study of the popular and much used Dirty Dozen, based on the largest and most diverse sample to date. All previous research results on the bi-factor structure and convergent validity were confirmed. Concerning the previous varying findings on one-factor, bi-factor, or three-factor solutions (cf. Carter et al., 2015), the large sample in the present study warrants to overall lean towards a bi-factor solution. However, a new (or old) problem was brought to the surface with the reporting of a strong bi-modal distribution of all three sub-factors, Machiavellianism, psychopathy, and narcissism. This has not been much emphasized in previous publications and a contribution of the present study is to highlight the scope of this problem and admonish for large samples sizes when researching the Dark Triad, which is known to compensate for unwanted distribution skewness in statistical analyses (cf. Lumley et al., 2002).

In the wake of the Mokken analysis by Carter et al. (2015), one of the attempts of this paper was to further the discussion on what the Dark Triad trait consists of and what the Dirty Dozen seeks to measure. First, the distribution analyses (Fig. 2) can be interpreted as an inertia to filling out Machiavellianism- and psychopathy-items, while narcissism-items showed normal, unskewed distribution, and consequently, not adding as much to the prediction of the core construct. A second clue to the latent core of the Dark Triad is the convergence analysis in Table 3 indicating that narcissism was the sub-factor that least correlated with the adjacent constructs of sub-factors to Mach-IV and Eysenck’s Personality Questionnaire, again showing that Machiavellianism-psychopathy is at the center of the construct. Although the rs in these convergence validity analyses were relatively low, if the convergent validity between two variables are modeled as loading on a latent ‘true’ variable, r = .50 would imply a whole 50% overlap (see Ozer, 1985; Trafimow, 2015). This could rightly be argued as a sufficient convergent validity. That being said, it is important to mention that there is another brief measure for assessment of the dark traits (i.e., the Short Dark Triad), which seems to perform better than the Dirty Dozen scale (e.g., Jones & Paulhus, 2014; Maples, Lamkin & Miller, 2014). In addition, although we used convergent analyses as in many other validation studies; recent research suggests that short scales should not be validated using these type of analysis (Olaru, Witthöft & Wilhelm, 2015). Third, the IRT-analysis showed that narcissism-items (e.g., need for admiration, attention, and status) had the least difficulty and the least discriminating power, not contributing to the total information on the latent dark trait. We conclude and submit for future research that the Dirty Dozen is a measurement consisting of a core found in Machiavellianism-psychopathy (e.g., manipulation, deceit, amorality, and callousness, with no remorse, as seen in Table 5).

Our proposal is that the Dark Triad, at least as measured by the Dirty Dozen, might be the product of a hasty grouping of two uncooperative sub-factors, Machiavellianism and psychopathy, together with one “desirable” sub-factor, narcissism (see also Garcia & Rosenberg, 2016). Being narcissistic is indeed considered more normal in these days and times (Twenge, Campbell & Freeman, 2012), not as undesirable to fill out in questionnaires, and does not add to the core of the construct. This suggests that grouping these three traits is unfortunate from both a social desirability (method artifact) and a subclinical perspective (how to find people with real problems). If one wants to quickly find the core of the Dark Triad, a one-item of “I exploit others” might be a suggestion. A similar approach has been taken recently with narcissism, compressing the original 40-item scale into a Single Item Narcissism Scale (SINS), and demonstrating reliability and validity in a number of studies (Konrath, Meier & Bushman, 2014; Van der Linden & Rosenthal, 2015).

Limitations and future research

The convergent validity analyses were limited to the fact that, outside the Dark Triad we only assessed extraversion and neuroticism. Both of these traits are related to affective experience and attention to emotional cues (Lucas, 2008). The inclusion of traits such as agreeableness, openness, and conscientiousness could be useful in future studies. That being said, one possible reason to the mixed results in earlier research is that these personality traits used to find differences or similarities between people’s dark character traits only represent individuals’ emotional reactions or temperament (e.g., McAdams, 2001; Haidt, 2006). After all, temperament is not useful in the distinction of who ends up with a mature or immature character (Cloninger, 2004). Indeed, not all individuals who are extroverts end up scoring high in psychopathy and/or narcissism (i.e., antecedent variables have different outcomes or “multi-finality”) and high scores in each one of the dark traits might have different antecedents (i.e., “equifinality”)” (see Cloninger & Zohar, 2011). Recent studies, for instance have found that the Dark Triad is rather a dyad when compared to Cloninger’s ternary model of “light” character traits: self-directedness, cooperativeness, and self-transcendence (Garcia & Rosenberg, 2016). Specifically, Machiavellianism and Psychopathy have a uncooperative and low self-directedness core, while narcissism is positively associated to self-directedness.

Although it was explicitly stated in the Introduction and Method sections that psychoticism, as measured in the Eysenck’s model, is better understood as the psychopathy trait in the Dark Triad, it is important to point out that most studies use other measures to operationalize this trait (e.g., the Self-Report Psychopathy Scale-II and III by Levenson, Kiehl & Fitzpatrick, 1995, respectively Paulhus, Neumann & Hare, in press). Nevertheless, other researchers have also used the measure developed by Eysenck to operationalize psychopathy in the Dark Triad (e.g., Linton & Power, 2013).

The varying social undesirability with all sub-factors measured by the Dirty Dozen should be further explored. It is not clear if the bi-modality of distribution is a reflection of this, or if it is a certain group of people with for instance very high Big Five-agreeableness, thus virtually hitting zero on all sub-factors of the Dirty Dozen. In other words, it is not clear if this indicates a genuine difference in the Dark Triad as a construct that is not instrument-specific. Additional studies have to be carried out using different methods and measures in order to assess whether or not such a difference is a method artifact or a real difference.

Another problem is that it is not apparent to what extent a short Dark Triad scale taps into and is confounded by clinical populations. In a large replication study such as the present, statistically 1–5% will be eligible for personality disorders. The results from IRT implicates that item-difficulties are sufficient on Machiavellianism and psychopathy to be able to distinguish problematic levels of the dark personality core, but not narcissism.

Conclusion

The conclusion on our part is that the Dirty Dozen has its advantages by being short, intuitive, and even fun, containing high face validity, but also has drawbacks by being highly differing in item-difficulties. The mismeasurement of the Dirty Dozen seems to be that it actually measures two constructs, narcissism and an anti-social trait. This specific conclusion is supported by the fact of what we choose to call a Single Item Dirty Dark Dyad (SIDDD), the “exploit” item. In situations of restrained research time and space, the SIDDD may capture the essence of what the Dirty Dozen actually measures.

Supplemental Information

Data S1 Dataset

Click here for additional data file.

Additional Information and Declarations

Competing Interests

Author Contributions

Human Ethics

Data Availability

1 “CrowdFlower is a data enrichment, data mining and crowdsourcing company based in the Mission District of San Francisco, California. The company’s software as a service platform allows users to access an online workforce of millions of people to clean, label and enrich data. CrowdFlower is typically used by data scientists at academic institutions, start-ups and large enterprises.” Retrieved from: https://en.wikipedia.org/wiki/CrowdFlower.

2 We are well aware of the controversy concerning composite scores present in the literature (e.g., Glenn & Sellbom, 2015). While we’re inclined to agree with their arguments on a theoretical level, we’ve elected to use a composite score as it is a quick abbreviation of a general “dark personality.” Furthermore, we conducted an exploratory omega analysis (function omega in R package psych, see also Revelle & Wilt, 2013), which yielded a ωhierarchical coefficient of .72, which suggests that the Dirty Dozen is saturated by a general factor. Additionally, the correlation between the composite score and an unrotated principal component was .994. This also serves as rationale for utilizing a unidimensional IRT model (see Ip, 2010). These analyses are available from the corresponding author upon request.

Danilo Garcia is the Director of the Blekinge Center of Competence, which is the Blekinge County Council’s research and development unit. Patricia Rosenberg is a researcher, project coordinator, and well-being coach at the Center. The Center works on innovations in public health and practice through interdisciplinary scientific research, person-centered methods, community projects, and the dissemination of knowledge in order to increase the quality of life of the habitants of the county of Blekinge, Sweden.

Petri J. Kajonius and Björn N. Persson analyzed the data, wrote the paper, prepared figures and/or tables, reviewed drafts of the paper.

Patricia Rosenberg wrote the paper, reviewed drafts of the paper.

Danilo Garcia conceived and designed the experiments, performed the experiments, analyzed the data, wrote the paper, prepared figures and/or tables, reviewed drafts of the paper.

The following information was supplied relating to ethical approvals (i.e., approving body and any reference numbers):

After consulting with the Network for Empowerment and Well-Being’s Review Board we arrived at the conclusion that the design of the present study (e.g., all participants’ data were anonymous and will not be used for commercial or other non-scientific purposes) required only informed consent from the participants.

The following information was supplied regarding data availability:

The raw data is supplied as Data S1.

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
