# Peer review of "The (mis)measurement of the Dark Triad Dirty Dozen: exploitation at the core of the scale"

_PeerJ, doi:10.7717/peerj.1748_

## Round 0.1 · original submission · Minor Revisions

· Academic Editor

Minor Revisions

The three reviewers have all published about the Dark Triad and offer constructive recommendations about your literature review, analyses (especially some lack of coherence between CFA and IRT), and conclusions. Please address the criticism of EPQ Psychoticism as a measure of psychopathy by justifying this choice as best you can even though you cannot change the choice now. Please address the criticism of your simplifying summary of the Dirty Dozen by the reviewers who mostly view it in terms of the Big Five, perhaps by reminding them that it can be seen in terms of different tests like the Big Five or the Psychobiological Model. A brief discussion of the single item conclusion to address the inconsistent comments of the reviewers will be helpful. Congratulations on a good paper that I expect to be accepted pending your addressing the reviewers consistent criticisms.

Reviewer 1 ·

Basic reporting

The manuscript, while mostly clear, could use some further attention to the general writing as some parts contain phrases that are a bit unclear or awkward (e.g., sentence spanning lines 380-384). The Introduction, which was longer than necessary, also missed several citations and studies that I would have expected to be cited. For instance, when discussing the validity of the Dirty Dozen (DD) scale that is the focus of this study, there was no mention of three studies that have been critical of this scale with regard to its convergent, discriminant, and criteria validity (e.g., with Big Five; e.g., Jones & Paulhus, 2014; Maples, Lamkin, & Miller, 2014; Miller et al., 2012). In particular, these studies originally demonstrated that the DD manifests quite limited convergent validity for psychological scales, that the DD Mach and Psychopathy scales manifest similar size correlations with established measures of these constructs (e.g., limited discriminant validity), and that the DD Narcissism scale has a different basic personality make-up (e.g. misses the high extraversion component that is found in most established measures of narcissism like the NPI). These studies should be cited given that they set the stage for critical examinations of the DD like the current one.

Other issues with the Introduction:
It doesn't make much sense to argue to me to argue that the Big Five focuses on light conceptualizations given that it has been used to understand psychopathy (e.g., Widiger & Lynam, 1998; Miller et al., 2001), DSM personality disorders (e.g., Lynam & Widiger, 2001; Samuel & Widiger, 2004), and the Dark Triad in general (e.g., O'Boyle et al., 2015). In fact, many trait PD theorists would argue that the DT, like other PDs (e.g., see DSM-5 section III) can be understood as being composed in large part of basic traits from the Big Five.

In terms of the measurement of psychopathy, I think it is inaccurate to say that Eysenck's psychoticism scale has been used for this purpose. This really isn't the case, certainly not very commonly in the psychopathy or DT literature. I don't understand why the authors chose to measure psychopathy in this more idiosyncratic manner, rather than using an established standalone measure as was done for Mach and narcissism. This was a methodological mistake in my mind. I don't think the authors should call the Eysenck psychoticism scale "psychopathy" in the manuscript (and tables) as I think this is misleading.

Experimental design

Other than the failure to include a validated measure of psychopathy, the methods generally seemed ok for the current purposes. I'd like to see them use some other methods to exclude individuals who may not have been attending to the stimuli or responding validly (other than excluding those who failed one of the two attention checks). Did they look to exclude individuals who used a preponderance of one answer response (e.g., all 1s or 2s) or were multivariate outliers?

I think the current study generally addresses a reasonably important issue, although there are several studies that have already examined several of these individual components. The authors are correct, however, in noting the impressively large sample their study brings to bear on these issues, which is particularly appropriate for several of the stats employed here (e.g., IRT).

Validity of the findings

While it is interesting to identify the one item that appeared to best capture the core of the DT, I think the authors should be more cautious in suggesting that this single item could be used as an assessment of the DT. I think there are very few situations that warrant the use of single item assessments of the DT or related constructs like narcissism as additional items can be added while only adding seconds to the length of the administration while increasing the reliability of the scores.

Also, in the Discussion I think there should be some general discussion of the limited convergent and discriminant validity found for the DD. In no other area of psychology, would we argue that a short measure that manifests convergent validity rs of .40 to .50 are sufficient. While brief measures are called for at times, it shouldn't be at the sake of messy or incomplete assessment. In fact, the authors should perhaps note that Jones and Paulhus have developed a brief measure of the DT (the Short Dark Triad), which is also an efficient measure that seems to perform better than the DD (Jones & Paulhus, 2014; Maples et al., 2014).

·

Basic reporting

The authors do a nice job of summarizing the state of affairs in the area of Dark Triad assessment.

Experimental design

The authors do an excellent job of reporting their design clearly and reporting transparent statistical procedures.

Validity of the findings

The findings are well described and are based on a large sample sufficient for the analyses conducted. The authors propose that a single item could capture the same amount of variance as most of the items, which is an interesting (and seemingly accurate) conclusion. The assertions appear completely in line with the data presented.

Comments for the author

There are a few minor points that could be fixed. For example, the authors have a typo where they say “narcissisms” I assume the authors mean narcissism. Also, the authors report convergent correlations of the Dirty Dozen with original measures from previous work (Jonason & Luevano, 2013) and state that the convergent correlations are “good” (.32-.53). I would disagree, a .32 correlation means you are capturing less than 10% of the original assessment, I would call that “poor.” Further, the authors should offer some further discussion over the finding that the Dirty Dozen subscales are mismatched with the original assessments.

Reviewer 3 ·

Basic reporting

In general the submission meets standards but I have a concern about some of the introduction.
The account of the DT traits offered in the introduction is overly simplistic. There exist broad literatures on both psychopathy and narcissism that are mostly ignored. Reviews of these literatures would reveal that neither psychopathy nor narcissism should be considered unidimensional traits in the manner conceived of here. From a general personality perspective, psychopathy consists of low levels of agreeableness and low levels of conscientiousness with a smattering of low neuroticism and high extraversion. At its core, narcissism consists of low levels of agreeableness and, depending on whether one is talking about the grandiose or vulnerable variant, high levels of extraversion or high levels of neuroticism.
It is a gross oversimplification when the authors write: “The Dark Triad embodies interpersonal, sub-clinical, and maladaptive personality traits in the general population (Paulhus & Williams, 2002), which are characterized by manipulativeness (i.e., Machiavellianism), impulsivity and antagonism (i.e., psychopathy), and the sense of entitlement (i.e., narcissism).” Manipulation is an aspect of psychopathy seen in most self-report psychopathy scales and in the gold standard Hare Psychopathy Checklist. Antagonism (i.e., low agreeableness) characterizes all three Dark Triad “traits.” Psychopathy is frequently considered to include grandiose narcissism as a component.

Experimental design

I had a cocern about one of the measures.
EPQ psychoticism is a poor measure of psychopathy. It is not used in the psychopathy literature at all. There are multiple self-report psychopathy measures available including the Hare SRP which is used in many DT studies. Why was one of these scales not used?

Validity of the findings

Most of my concerns were regarding the specific analyses.

The authors test two models using CFA—a single factor model in which all items are indicators of the overall factor and a bi-factor model in which a general factor is modelled along with three specific factors. I did not understand why only two models were tested in the CFA. Some researchers contend that Machiavellianism is a subset of psychopathy. Such a model suggests a two-factor model. Why not be more empirically driven and conduct an exploratory analysis on half of the sample and compare that model to the a priori ones?

It is not surprising that the single composite DD DT score correlates more strongly with the individual DD components than with the stand alone measures. The single DD components make up the composite—these are part-whole correlations. This should be dropped.

Related to the point above, the discussion of intercorrelations between the DD single scales and the strand alone scales should also mention discriminant validity. It appears that DD psychopathy correlates more strongly with stand-alone Machiavellianism than it does with EPQ psychoticism.
6. The authors indicate that the three DD scales are bi-modal and suggest that this may be due to social desirability in the items. This seems quite speculative. Could it also represent some degree of nay-saying, acquiescence, or the existence of a subpopulation in the sample? Is there any way to bring data to bear on this? Can the authors code items according to their degree of social desirability?

I do not understand why the authors model all 12 items in the IRT analysis when results from the previous CFA suggest that there are three separate factors. That is,how can the authors conclude that there is adequate unidimensionality among the 12 items for the IRT analyses but conclude they are multidimensional in the CFA analyses?

---

## Round 0.2 · accepted · Accept

· Academic Editor

Accept

Thank you for an excellent revision. The paper is a valuable contribution to the literature.